# Water Cycle Health Assessment Using the Combined Weights and Relative Preference Relationship VIKOR Model: A Case Study in the Zheng-Bian-Luo Region, Henan Province

**Mengdie Zhao [1,2,3], Jinhai Wei [2], Yuping Han [2,3,*] and Jinhang Li [2]**

[1] Yellow River Survey Planning and Design Institute Co., Ltd., Zhengzhou 450046, China; zhaomengdie@ncwu.edu.cn

[2] College of Water Resources, North China University of Water Resources and Electric Power, Zhengzhou 450046, China; z202210010173@stu.ncwu.edu.cn (J.W.); 201703721@stu.ncwu.edu.cn (J.L.)

[3] Key Laboratory of Intensive Conservation of the Yellow River Basin, Zhengzhou 450046, China

\* Correspondence: comhanyp@ncwu.edu.cn

**Abstract:** Both the natural and social water cycles form part of the regional water cycle, and the assessment of the health of the social water cycle provides useful recommendations for resource allocation, urban planning, and development. The Zheng-Bian-Luo region (Zhengzhou, Kaifeng, Luoyang city cluster in China) is used as an example in this study. The three-level "goal criterion index" is used to develop a water cycle index system based on deeper knowledge of the notion of the social water cycle. The system has four criterion layers that measure water quantity, utility, quality, and ecology, in addition to 22 index levels regarding the total water resources and drinking water compliance rate. By using this as a foundation, the minimum information entropy principle was applied to couple AHP (Analytic Hierarchy Process) and EFAST (Extended Fourier Amplitude Sensitivity Analysis) in order to calculate the comprehensive weights of the evaluation indicators and build a VIKOR (Intuitionistic Fuzzy Multi-attribute Decision Making Method) model of the relative preference relationship of the fused weights. This model was then compared to the conventional VIKOR model and the FCE (Fuzzy Comprehensive Evaluation Method) method in order to reflect on the objectivity of the evaluation results. The primary barriers preventing the improvement of water cycle health in the Zheng-Bian-Luo region were determined in this study by using the barrier degree model. The findings demonstrate that over the past 11 years, the overall water cycle health in the Zheng-Bian-Luo region has developed toward a healthy trend and that the water cycle health level in the region has gradually improved from the initial sub-pathological state to a healthy state. The results also demonstrate compliance with domestic drinking water sources, comprehensive water consumption per capita, the water consumption of CNY 10,000 of industrial-added value, the water consumption of CNY 10,000 of GDP, and the water consumption of CNY 10,000 for water. The primary barrier to the Zheng-Bian-Luo region's improvement in water health is the water consumption ratio. The findings of this study can serve as a scientific foundation for creating a balanced urban water cycle and achieving long-term development in the area.

**Keywords:** healthy water cycle; combined weights; VIKOR model; barrier factor analysis; Zheng-Bian-Luo

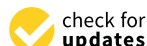



## 1. Introduction

The urban water cycle has gradually evolved into a complex "natural-social" dual water cycle process involving the natural processes of rainfall, runoff, and evaporation, and the social processes of water intake, water supply, water use, and drainage as social economies have grown, with human activities have become more intense [1,2]. However, the pathological water cycle has negatively impacted human life and the environment, making it difficult to manage water resources sustainably. As a result, a sound and functional

urban water cycle system (Figure 1) is crucial to the sustainability of the local biological environment and quality of life [3].

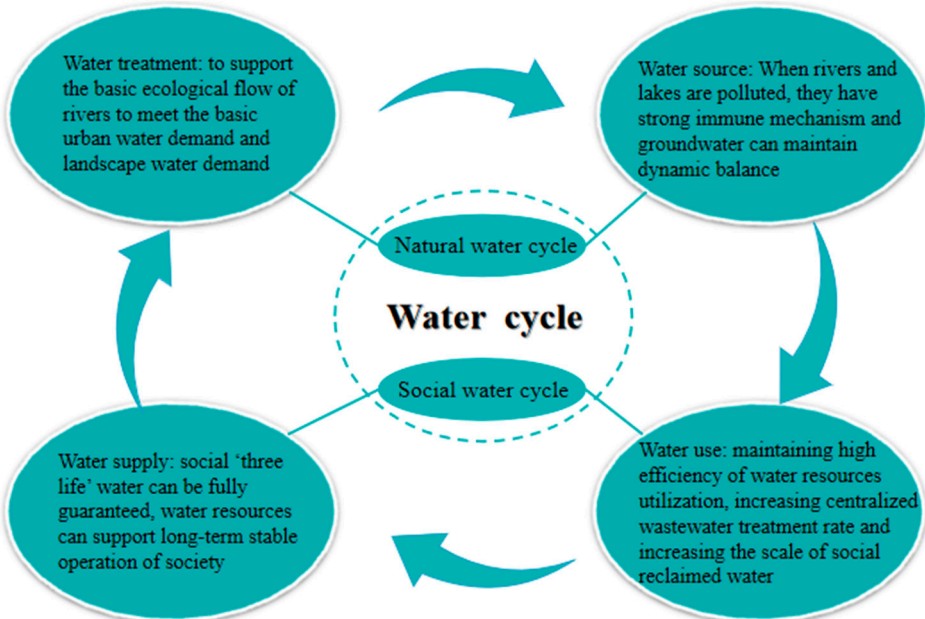

**Figure 1.** Healthy urban water cycle schematic.

Numerous domestic specialists and academics started researching the health of the urban water cycle based on the general trend of water resources, water ecology, and water environment as the problem of water resources became more and more apparent. Zhang J. et al. [4] first proposed the concept of a healthy water cycle, mainly emphasizing the use of recycled water and the popularization of wastewater purification as the key to a healthy water cycle. Xu Xiangjun et al. [5] examined the idea of an urban water cycle and suggested ways to build one in accordance with the elements that influence it. Some researchers have used the binary characteristics of water resources to build an assessment index system. For example, Tang Jizhang et al. [6] studied the health of the water cycle in Xi'an based on the principle of "target criteria indicator", which is based on the attributes of the binary water cycle in the city. The results show that improving the rationality of water resources development and utilization and alleviating the contradiction between the supply and demand of water resources is the key to improving the health of the water cycle in Xi 'an. Some experts and scholars evaluate the water cycle health of cities or regions based on the coupling of the natural and social aspects of the water cycle. Wang [7] evaluated the health of the water cycle of farmland from four dimensions: water source, water extraction and transmission subprocess, water consumption subprocess, and drainage retreat subprocess. The results show that the health degree of farmland in the irrigation area is developing towards a good trend after a series of measures have been taken by human beings. Chen Jiongli [8] gave an evaluation of the water cycle in Yinchuan City from four dimensions, including the water ecology level, water environment quality, water resource abundance, and water resource utilization. The results show that the differences in each dimension are small and are developing towards a good trend, and it is concluded that the amount of sewage regeneration is the main index affecting the health of the water cycle in Yinchuan. Ma Jing et al. [9] evaluated the water cycle from five perspectives: water source, water supply, water use, drainage, and reuse. The results show that although each dimension has some fluctuations, the overall trend is improving. It also shows that the adjustment of industrial structure and the implementation of water conservancy and people's livelihood policies in Handan City are important factors affecting the health of the water cycle in Handan City.

Other specialists and academics have examined the health of the water cycle using a variety of evaluation techniques and views based on their own expertise. Yang Haiyan et al. [10] evaluated the water-carrying capacity of Weifang City based on the VIKOR method (Intuitionistic Fuzzy Multi-attribute Decision Making Method), and the results showed that the water-carrying capacity of some areas in Weifang City did not match the local water conditions. Li Yinjiu et al. [11] evaluated the health degree of the Guangdong River based on a composite fuzzy matter element VIKOR model. The study showed that the quality of the river water was the main factor affecting the health level of the river. Li Na et al. [12] evaluated the sustainability of packaging schemes based on the entropy weight-VIKOR model. The results show that the packaging scheme with paperboard as the main material has the best sustainability. He Gang et al. [13] evaluated the water and soil ecological security of mining cities in Anhui Province based on the VIKOR model. The results show that the results obtained by VIKOR are in good agreement with the actual results, and the per capita water consumption is the main factor affecting the water and soil ecological security of mining cities in Anhui Province. Wang Lunyan et al. [14] evaluated the water resources carrying capacity of nine provinces and regions in the Yellow River Basin based on a fuzzy set-pair analysis method with combined weights and a barrier degree model. The study showed that the overall improvement trend of the carrying capacity of the Yellow River Basin provinces was obvious. Bai Fangfang et al. [15] objectively evaluated the utilization efficiency of agricultural water resources in nine provinces and regions in the Yellow River Basin based on the entropy weight TOPSIS model (Technique for Order Preference by Similarity to an Ideal Solution). The study showed that the overall agricultural water resource utilization efficiency of each province improved, and the gap in agricultural water resource utilization efficiency between the provinces became significantly smaller. Despite the fact that the water cycle has been researched in other nations in the past, there has not been as much research carried out regarding its health, and only a few academics have looked into the best ways to assess this. Gani et al. [16] used distance estimation to govern the urban water cycle. Deng et al. [17] gave an evaluation of the health of the Taihu Lake Basin in China based on an improved entropic fuzzy material element model. Meneses et al. [18] evaluated the urban water cycle based on the life cycle assessment method, revealing that the non-drinking water use of reclaimed water in the urban water cycle has environmental and economic advantages. Pinto et al. [19] evaluated river health based on factor analysis. According to the study, eutrophication, microbiological contamination, and anaerobic fermentation are the key issues that have an impact on the health of rivers.

There are few studies on the comprehensive evaluation of water cycle health in urban agglomerations, and few studies analyze the water cycle health of urban agglomerations from the perspective of time and space. A healthy water circulation system is the premise and foundation for the high-quality development of urban agglomerations, and the evaluation of water circulation health is the key link to improving the health of urban agglomerations. Therefore, the evaluation of the current water resources situation and urban development model of urban agglomerations can provide a theoretical basis for the sustainable utilization of water resources and regional sustainable development of urban agglomerations.

Based on the foregoing context, this paper uses the Zheng-Bian-Luo area (Zhengzhou, Kaifeng, Luoyang city cluster in China)as its research object and develops four criterion layers: water abundance (A), water utility (B), water quality (C), and water ecology (D), including the domestic water consumption rate, groundwater water supply ratio, water production coefficient, etc., within a total of 22 indicators of the water cycle health evaluation index system. By using the idea of least information entropy, it was possible to combine the subjective and objective index weights that were produced from the subjective weights derived from AHP (Analytic Hierarchy Process), with the objective weights derived from the EFAST (Extended Fourier Amplitude Sensitivity Analysis) algorithm. The VIKOR model of the relative preference relationship of combined weights is built on the basis of weight calculations. The water cycle health evaluation of the Zheng-Bian-Luo area was

conducted from 2011 to 2021. The VIKOR model was then used to compare and verify using FCE (Fuzzy Comprehensive Evaluation Method), and finally, the barrier degree model was introduced to analyze the barrier factor of the Zheng-Bian-Luo area.

In the selection of methods, we know that the essence of subjective weighting methods such as AHP and ANP (Analytic Network Process) is that decision-makers subjectively determine the weight of each index based on experience. Although its explanation is strong, its objectivity is poor. The essence of objective assignment methods such as the entropy weight method and EFAST method is that the original data for calculating weights are obtained from the actual data of evaluation indicators in the process of evaluation. Although the objective assignment method to determine the weight accuracy is higher, it is contrary to the actual situation and has a poor interpretation. Therefore, subjective and objective comprehensive weights are adopted. That is the subjective assignment method AHP and the objective assignment method EFAST.

The findings of this study, which aimed to better understand water cycle health and provide a theoretical underpinning for the sustainable development of Henan Province, show, on the one hand, the factors that act as barriers to water cycle health (in various urban areas) and the spatial and temporal distribution and evolution of water cycle health in the Zheng-Bian-Luo region on the other.

The innovation of this study lies in the process of constructing the index system, we consider that the extraordinarily large flood in Zhengzhou on July 20, 2021, has a significant impact on the health of the water cycle. Therefore, we add annual precipitation to the construction of the evaluation index system, which is innovative compared with previous studies. Secondly, in the evaluation of innovative methods, the relative preference relationship of fusion weights is constructed to replace the aggregation function $L_P - metric$ to improve the VIKOR model.

The composition of this article is mainly divided into the following four parts, and their respective roles are not the same. Introduction: Introduce the background of the research problem, the problem, and the selection of methods. Materials and Methods: This paper introduces detailed information on Zheng-Bian-Luo urban agglomeration in the study area, as well as the selection of indicators, the search for data, and the specific operation of the method. Results and Discussion: The results calculated by the method are analyzed from three perspectives: target layer, dimension layer, and index layer, and then the obstacle degree model is used for analysis, and finally compared with other methods. Conclusions: Based on the above analysis, draw conclusions.

## 2. Materials and Methods

### 2.1. Study Area

The three cities of Zhengzhou, Kaifeng, and Luoyang in Henan Province are referred to as Zheng-Bian-Luo [20,21]. The three regions are arranged in an "L" pattern along the Yellow River, which serves as the cultural center of the Central Plains (Figure 2). There are four different seasons in the Zheng-Bian-Luo region's north temperate continental monsoon climate. The majority of Zheng-Bian-Luo is a semi-arid and semi-humid region with inconsistent rainfall distribution, wide economic development gaps, poor water resource utilization rates, a lack of water resources, and weak water cycle health. As a result, the assessment of the health of the water cycle in the Zheng-Bian-Luo region may serve as a foundation for improving the state of the water environment there, as well as the cycle's overall health.

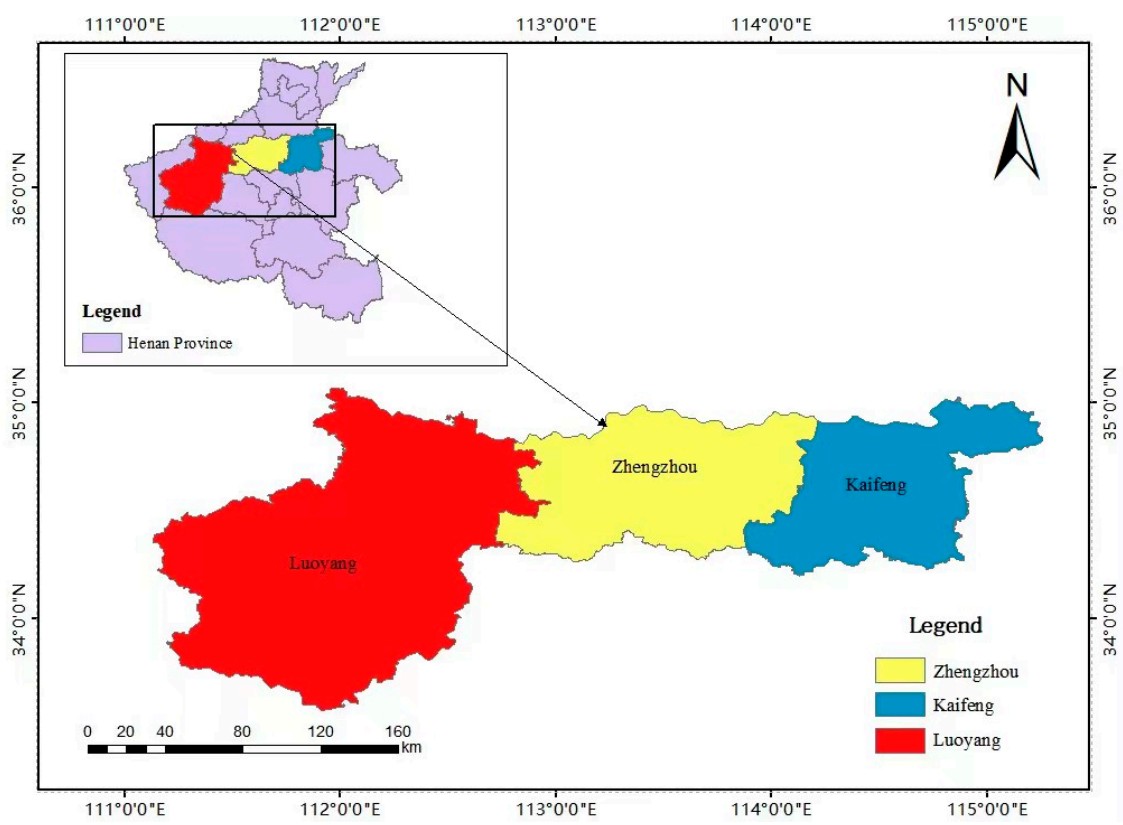

**Figure 2.** A map of the Zheng-Bian-Luo region.

### 2.2. Data Source

A total of 11 years of raw data from 2011 to 2021 for the Zheng-Bian-Luo area was gathered through the collection and analysis of pertinent data from China's Water Resources Bulletin, China's Statistical Yearbook, and China's Environmental Bulletin [22].

### 2.3. Establishment of the Evaluation Index System

Both natural and social water cycle processes must be taken into account when evaluating the health of the urban water cycle. The choice of indicators is crucial for performing the evaluation, and if the choice is poor, the health of the urban water cycle cannot be accurately and fully reflected [23,24]. As a result, while choosing the indicators, we should adhere to the criteria of objectivity while also taking into account regional characteristics. In this paper, we establish the water cycle health evaluation system from four dimensions: water abundance, water utility, water quality, and water ecology. We do this by carefully considering the connotation of a healthy urban water cycle and the current situation of the region and by collecting pertinent basic data for the Zheng-Bian-Luo region by consulting the Henan Statistical Yearbook and other related materials. The primary factors taken into account in the water abundance criterion layer are the natural circumstances under which urban water resources are naturally endowed, such as the total volume of water resources, the percentage of groundwater supplies, etc. The water utility index layer is primarily based on the effective utilization coefficient of irrigation water and the social "three activities" to produce the index layer, representing the effectiveness of how well urban water resources are used. The level of drinking water standards, the degree of urban greening, and the condition of water-functional regions are the key factors that the water quality dimension layer considers when reflecting the natural characteristics of the water cycle. The human protection of ecology, the storage capacity of rivers and lakes, the density of drainage pipes, and the amount of change in the shallow groundwater level are the primary components of the water ecology dimension layer. (In short, in the evaluation

system, the selection of the index layer mainly follows two criteria. Firstly, according to the existing research results, the indicators with a high correlation with water cycle health are selected, which can make the selected indicators scientific. Second, according to the characteristics of the study area, select regional representative indicators, so that the selected indicators are independent and representative). The evaluation index system is shown in Figure 3.

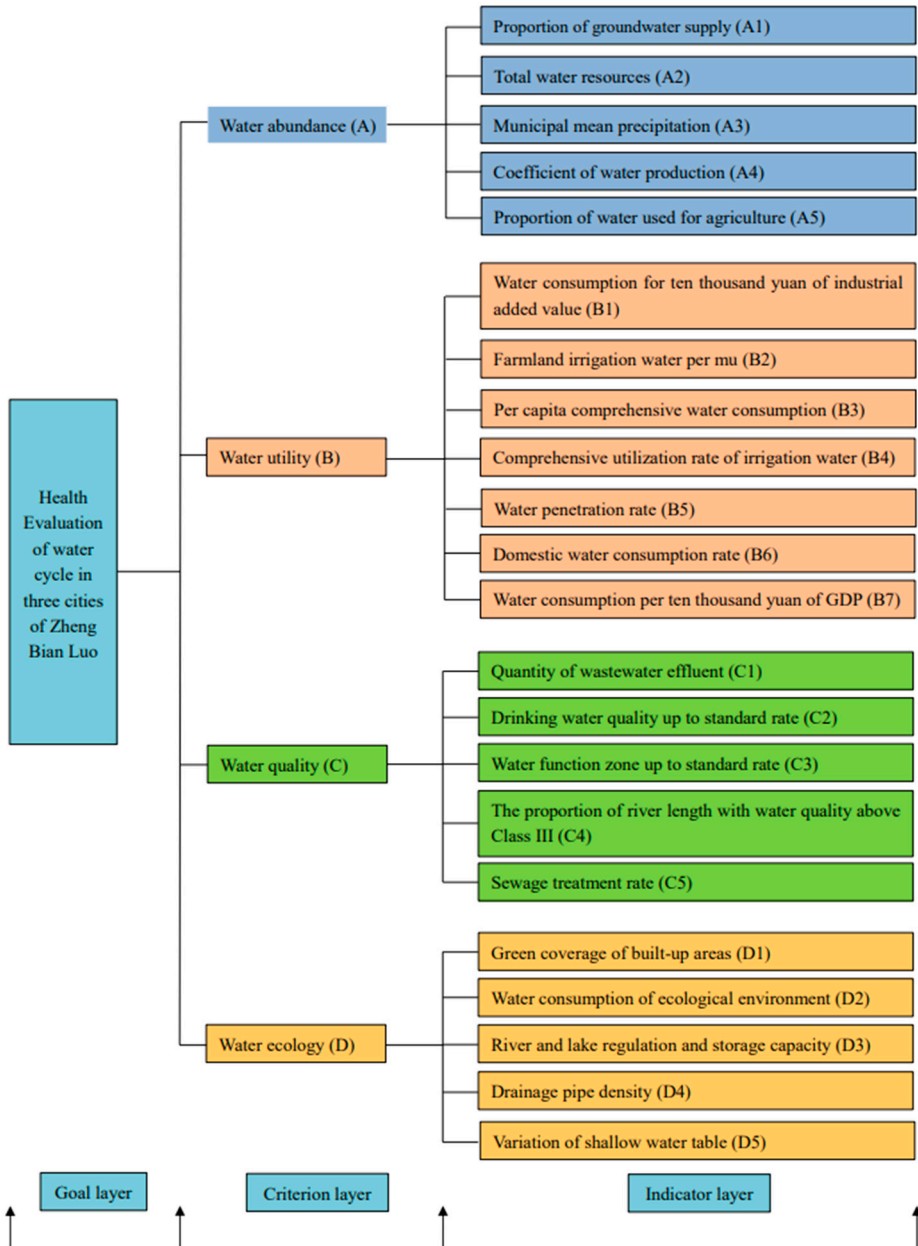

**Figure 3.** System for evaluating the health of the water cycle.

Evaluation Indicator Threshold Standard

The water cycle health grade was divided into five levels [1,25], namely healthy (level I), sub-healthy (level II), average (level III), sub-pathological (level IV), and pathological (level V). Surveys, references to existing relevant information, and national norms were used to determine the thresholds related to the water resource health assessment. Table 1 displays the index properties as well as the assessment system's grade and thresholds.

**Table 1.** Grading standards and evaluation index qualities.

| Criterion Layer | Index Layer | I | II | III | IV | V | Attribute |
|---|---|---|---|---|---|---|---|
| | | 5 | (5,4] | (4,3] | (3,2] | (2,1] | |
| Water abundance A | A1/% | [10,25) | [25,40) | [40,55) | [55,70) | [70,100) | Negative |
| | A2/108 × m$^3$ | >10 | [10,8) | [8,6) | [6,4) | ≤4 | Positive |
| | A3/mm | [650,600) | [600,550) | [550,500) | [500,450) | ≤450 | Positive |
| | A4 | [0.5,0.4) | [0.4,0.3) | [0.3,0.2) | [0.2,0.1) | ≤0.1 | Positive |
| | A5/% | [10,25) | [25,40) | [40,55) | [55,70) | [70,100) | Negative |
| Water utility B | B1/m$^3$ | [10,25) | [25,50) | [50,100) | [100,150) | ≥150 | Negative |
| | B2/m$^3$ | ≥150 | (150,100] | (100,80] | (80,50] | ≤5 | Positive |
| | B3/m$^3$ | >200 | [200,150) | [150,90) | [90,50) | ≤50 | Positive |
| | B4 | [0.85,0.75) | [0.75,0.65) | [0.65,0.55) | [0.55,0.45) | [0.45,0) | Positive |
| | B5/% | [100,95) | [95,90) | [90,85) | [85,80) | ≤80 | Positive |
| | B6/% | ≤10 | (10,20] | (20,30] | (30,50] | >50 | Negative |
| | B7/(m3/CNY 10,000) | [10,25) | [25,50) | [50,100) | [100,150) | ≥150 | Negative |
| Water quality C | C1/108 × m$^3$ | 5 | (5,4] | (4,3] | (3,2] | (2,1] | Negative |
| | C2/% | 100 | (100,95] | (95,90] | (90,80] | <80 | Positive |
| | C3/% | [100,90) | [95,60) | [60,40) | [40,20) | [20,0) | Positive |
| | C4/% | [100,60) | [60,50) | [50,30) | [30,20) | [20,0) | Positive |
| | C5/% | [100,95) | [95,90) | [90,85) | [85,80) | ≤80 | Positive |
| Water ecology D | D1/% | [100,50) | [50,40) | [40,30) | [30,20) | [20,10) | Positive |
| | D2/108 × m$^3$ | [10,5) | [5,4) | [4,3) | [3,2) | [2,0) | Positive |
| | D3 | Strong | Stronger | Fair | Weaker | Weak | Positive |
| | D4/km km$^{-2}$ | >15 | [15,12) | [12,9) | [9,6) | ≤6 | Positive |
| | D5/m | ≤−2 | (−2,−1] | (−1,0] | (0,1] | >1 | Positive |

*2.4. Research Methods*

2.4.1. AHP-EFAST Algorithm

The objective weights were obtained using the EFAST algorithm, while the subjective weights were obtained using the AHP algorithm. By using the principle of minimum information entropy, the combined weights were then obtained by combining the combined subjective and objective weights. This process aims to produce accurate and reasonable weights for each index. These are the calculation stages [26,27].

Step1: Standardization of data.

In order to remove the impacts brought on by extreme values, which were created by the once-in-a-millennium unusual rainfall that hit Zhengzhou City in 2021 and left behind unrepresentative extreme value data, the standardized change approach is utilized in this article.

$$r_{ij} = (x_{ij} - X_j)/S_j \qquad (1)$$

where $x_{ij}$ is the evaluation index's initial value; $r_{ij}$ is the index's value, following the standard transformation procedure; $Xj$ is the $j$th index's mean value; and $S_j$ is its standard deviation.

Step2: AHP determines subjective weights.

a: Constructing a simple hierarchy for the water cycle;

b: Create the U-judgment matrix.

$$U = \begin{bmatrix} U_{11} & \cdots & U_{1n} \\ \vdots & \vdots & \vdots \\ U_{1n} & \cdots & U_{nn} \end{bmatrix} \qquad (2)$$

c: Determine each indicator's weight $W_j'$

$$UW = \lambda_{\max} W \tag{3}$$

$$W_j' = \frac{\sqrt[n]{\prod_{j=1}^{n} U_{ij}}}{\sum_{i=1}^{n} \sqrt[n]{\prod_{j=1}^{n} U_{ij}}} \tag{4}$$

d: Test for consistency. If $CR > 0.1$, the consistency test fails, and a new reconstruction of the judgment matrix must be carried out until CR0.1. The formula for calculating this is

$$CR = CI/RI \tag{5}$$

$$CI = \frac{1}{n-1}(\lambda_{\max} - n) \tag{6}$$

where $RI$, which is determined by the order $n$ of the matrix, is the randomness index. See Table 2 below.

**Table 2.** Values of the consistency test RI.

| Order | 1 or 2 | 3 | 4 | 5 | 6 | 7 | 8 | 9 |
|---|---|---|---|---|---|---|---|---|
| R.I. | 0 | 0.52 | 0.89 | 1.12 | 1.26 | 1.36 | 1.41 | 1.46 |

Step3: The EFAST algorithm determines the objective weights.

The whole sensitivity analysis approach used by the EFAST algorithm is variance decomposition-based, meaning that the variance resulting from the indicators' mutual coupling reflects the sensitivity. The following is the EFAST algorithm.

The sum of the variances of the coupling effects among the indicators may be used to indicate the overall variance of the model's output:

$$V = \sum_i V_i + \sum_{i \neq j} V_{ij} + \sum_{i \neq j \neq k} V_{ijk} + \cdots + \sum V_{ijk \cdots n} \tag{7}$$

where $V$ is the overall variance of the model's output; $V_i$ is the variance of the indicator $x_i$; $V_{ij}$ is the variance of the interaction between the indicators $x_i$ and $x_j$; $V_{ijk}$ is the variance of the interaction between the indicators $x_i$, $x_j$, and $x_k$, and $x_{ijk\,n}$ is the variance of the interaction of the indicator $x_i$ through the remaining $n - 1$ indicators.

$T_i$ the first-order sensitivity index, denoting the direct contribution of a single indicator $x_i$ to the total variance of the model's output, the second-order $T_{ij}$, the third-order $T_{ijk}$, and the higher-order sensitivity index $T_{ijkn}$ of the coupling effect of indicator $x_i$, along with other indicators, can be defined as

$$T_i = \frac{V_i}{V}, \; T_{ij} = \frac{V_{ij}}{V}, \; T_{ijk} = \frac{V_{ijk}}{V}, \; T_{ijk \cdots n} = \frac{V_{ijk \cdots n}}{V} \tag{8}$$

The total of each rank of indicator sensitivity is

$$T_{ti} = T_i + T_{ij} + T_{ijk} + T_{ijk \cdots n} \tag{9}$$

To make the weights more thorough, after considering the sensitivity of the indicators, the coupling impact, and the weights to be obtained, normalization was used to produce $W_j''$. Different indicators have various effects on the model outcomes. The following is the formula:

$$W_j'' = T_{ti} / \sum_{i=1}^{n} T_{ti} \tag{10}$$

Step4: Determine the subject-objective combination of the weight of the indicators:

$$W_j = \frac{W_j' * W_j''}{\sum_{j=1}^{n} W_j' * W_j''} \tag{11}$$

where $W_j$ is the weight value for the indicator combination, $n$ is the number of indicators, $W_j'$ is the subjective weight result produced from AHP, and $W_j''$ is the objective weight result derived from the EFAST algorithm.

2.4.2. Upgraded VIKOR Technique

By taking into consideration the subjective preferences of the decision maker, the VIKOR technique is able to balance the "individual regret value" with the "group benefit value", therefore obtaining the best compromise option that the decision-maker can compromise [28,29]. The distance between the assessed solution and the ideal solution is calculated using the conventional VIKOR approach using the $L_p - metric$ aggregation function, specifically:

$$L_{p-metric} = \left\{ \sum_{i=1}^{n} \left[ W_j (y_0^+ - y_{ij}) / (y_0^+ - y_0^-) \right]^p \right\}^{1/p} \tag{12}$$

where $L_p - metric$ is the usual distance between the thing to be judged and the ideal answer, and the lower its value, the better the answer; $y_{ij}$ stands for the standard data; the perfect solutions $y_0^+$ and $y_0^-$, respectively; $p$ stands for the aggregation function calculation coefficient.

By employing linear weighting, the normalized output of the $L_p - metric$ aggregation function is produced; however, this result deviates significantly from reality. In order to replace the $L_p - metric$ aggregation function, it is necessary to do an analysis of relative preference relationships. Let $\tilde{c} = (c_1, c_2, c_3)$ and $\tilde{d} = (d_1, d_2, d_3)$ be two triangular fuzzy numbers, and the relative preference relationship operator $V_{p*}$ between $\tilde{c}$ and $\tilde{d}$ is defined as follows:

$$
\begin{aligned}
V_{p*}(\tilde{c}, \tilde{d}) &= \frac{1}{2} \left( \frac{(c_1 - d_3) + 2(c_2 - d_2) + (c_3 - d_1)}{2\|M\|} \right) \\
\|M\| &= \begin{cases} \frac{(m_1^+ - m_3^-) + 2(m_2^+ - m_2^1) + (m_3^+ - m_1^-)}{2} & if : m_1^+ \geq m_3^- \\ \frac{(m_1^+ - m_3^-) + 2(m_2^+ - m_2^1) + (m_3^+ - m_1^-)}{2} + 2(m_3^- - m_1^+) & if : m_1^+ < m_3^- \end{cases} \\
m_1^+ &= \max\{c_1, d_1\}, \qquad m_2^+ = \max\{c_2, d_2\}, \qquad m_3^+ = \max\{c_3, d_3\}, \\
m_1^- &= \max\{c_1, d_1\}, \qquad m_2^- = \max\{c_2, d_2\}, \; m_3^- = \max\{c_3, d_3\}
\end{aligned} \tag{13}
$$

The preference for fuzzy number $\tilde{c}$ over fuzzy number $\tilde{d}$ is indicated by the letter $V_{p*}(a, b)$. $V_{p*}(a, b) > 0.5$, $\tilde{c}$ has a higher preference degree than $\tilde{d}$. The preference degree of $\tilde{c}$ is the same as $\tilde{d}$'s if $V_{p*}(a, b) = 0.5$. When $V_{p*}(a, b) > 0.5$, $\tilde{c}$ has a higher preference degree, making it superior to $\tilde{d}$. Conversely, $\tilde{d}$ has a higher preference degree than $\tilde{c}$. If $A = 0.5$, then $\tilde{c}$ and $\tilde{d}$ have the same preference degree.

According to this analysis, when the value of $V_{p*}$ is equal to 0.5, the deviation of $\tilde{c}$ from $\tilde{d}$ is zero. Conversely, when the value of $V_{p*}$ is further away from 0.5, the departure of $\tilde{c}$ from $\tilde{d}$ is smaller. As a result, the amount of the distance between the triangular fuzzy

numbers $\tilde{c}$ and $\tilde{d}$ may be expressed using the value of $V_{p^*}$. So, the formula for the distance between $\tilde{c}$ and $\tilde{d}$ is

$$l(\tilde{c}, \tilde{d}) = \begin{cases} V_{p^*}\left(\tilde{c}, \tilde{d}\right) - 0.5 \, if : \tilde{c} \geq \tilde{d} \\ 0.5 - V_{p^*}\left(\tilde{c}, \tilde{d}\right) if : \tilde{c} \leq \tilde{d} \end{cases} \tag{14}$$

Step1: Attribute standardization.

For benefit-based metrics:

$$r_{ij} = \frac{a_{ij}}{\sqrt{\sum\limits_{i=1}^{m} a_{ij}^2}} i = 1, 2, \cdots\cdots m, j = 1, 2, \cdots\cdots n \tag{15}$$

For cost-based metrics:

$$r_{ij} = \frac{\frac{1}{a_{ij}}}{\sqrt{\sum\limits_{i=1}^{m} \left(\frac{1}{a_{ij}}\right)^2}} i = 1, 2, \cdots\cdots m, j = 1, 2, \cdots\cdots n \tag{16}$$

where $r_{ij}$ is the $j$th evaluation index's standardized data from the $i$th program, and $a_{ij}$ denotes the index's original data from the same program.

Step2: Based on the normalized data, identify the fuzzy ideal solutions $\tilde{z}_j^+$ and $\tilde{z}_j^-$.

$$\tilde{z}_j^+ = \left\{ \left(\max_{1 \leq i \leq n}(r_{ij})|j \in J_1\right), \left(\min_{1 \leq i \leq n}(r_{ij})|j \in J_2\right) \right\} \tag{17}$$

$$\tilde{z}_j^- = \left\{ \left(\min_{1 \leq i \leq n}(r_{ij})|j \in J_1\right), \left(\max_{1 \leq i \leq n}(r_{ij})|j \in J_2\right) \right\} \tag{18}$$

where $J_1$ is the collection of data based on benefits, and $J_2$ is the collection of indicators based on costs.

Step3: Calculate the group benefit value $S_i$ and the individual regret value $R_i$.

$$S_i = \sum_{j=1}^{n} \frac{W_j d(\tilde{z}_j^+, r_{ij})}{d(\tilde{z}_j^+, \tilde{z}_j^-)} \tag{19}$$

$$R_i = \max_{j} \frac{W_j d(\tilde{z}_j^+, r_{ij})}{d(\tilde{z}_j^+, \tilde{z}_j^-)} \tag{20}$$

Maximum individual regret value $R_i$ and group benefit $S_i$ are calculated using $d(\tilde{z}_j^+, r_{ij})$ as the index distance and $d(\tilde{z}_j^+, \tilde{z}_j^-)$ as the standard distance, respectively. The information in the original data will not be properly used if the triangular fuzzy numbers are immediately anti-fuzzy-translated into exact values to compute their distances. As a result, this study uses the relative preference relation operator $V_{p^*}$ to indirectly address the aforementioned problem. The revised equation is as follows:

$$S_i = \sum_{j=1}^{n} \frac{W_j\left[V_{p^*}(z_j^+ - \tilde{r}_{ij}) - 0.5\right]}{V_{p^*}(z_j^+ - \tilde{z}_j^-) - 0.5} \tag{21}$$

$$R_i = \max_j \frac{W_j \left[ V_{p^*}(z_j^+ - \widetilde{r}_{ij}) - 0.5 \right]}{V_{p^*}(z_j^+ - \widetilde{z}_j^-)^i - 0.5} \tag{22}$$

Step 4: Calculate each evaluation index's total evaluation value.

$$Q_i = v\frac{S_i - S^-}{S^+ - S^-} + (1-v)\frac{R_i - R^-}{R^+ - R^-}$$
$$S^+ = \max S_i, \ S^- = \min S_i, \ R^+ = \max R_i, \ R^- = \min R_i \tag{23}$$

$v$ is the coefficient of the maximum group utility decision strategy; when $v$ is less than 0.5, it means that the strategy is formulated in a way that minimizes individual regrets with a greater proportion; when $v$ is equal to 0.5, it means that the strategy maximizes group benefits while minimizing individual regrets, and when $v$ is greater than 0.5, it means that the strategy is formulated in a way that maximizes group benefits with a greater proportion. $v \approx 0.5$ is employed in this paper.

Step 5: Order the evaluation initiatives.

From smallest to largest, arrange $Q_i$, $S_i$, and $R_i$. The better the plan and the higher the degree of health, the smaller the value of $Q_i$. According to the equation above, $Q_i$ stands for water cycle health and has a value between 0 and 1. The dewatering cycle is in better shape the lower the value.

2.4.3. Obstacle Factor Analysis

Step 1: Determine the $j$th evaluation index of $F_j$'s contribution.

$$F_j = R_j \times W_i \tag{24}$$

where $R_j$ is the categorization indication's weight for the $i$th indicator. $W_i$ is the combined indicator's weight of it.

Step 2: Calculation of deviation $I_j$:

$$I_j = 1 - x_{ij} \tag{25}$$

Step 3: Determine each indicator's barrier degree $P_j$.

$$P_j = \frac{F_j I_j}{\sum_{j=1}^n F_j I_j} \times 100\% \tag{26}$$

According to the aforesaid methodology, the barrier degree factor analysis of all items can determine the barrier degree of each individual indication. Table 3 is a list of abbreviations and variables used. Figure 4 depicts the flow chart used to assess the health of the water cycle.

**Table 3.** A list of abbreviations and variables used.

| Variable | Interpretation | Variable | Interpretation |
|---|---|---|---|
| $x_{ij}$ | the evaluation index's initial value | $r_{ij}$ | the index's value |
| $Xj$ | the $j$th index's mean value | $S_j$ | its standard deviation. |
| $V$ | the overall variance of the model's output | $V_i$ | the variance of the indicator $x_i$ |
| $V_{ij}$ | the variance of the interaction between the indicators $x_i$ and $x_j$ | $V_{ijk}$ | the variance of the interaction between the indicators $x_i$, $x_j$, and $x_k$ |
| $V_{ijk\,n}$ | the variance of the interaction of the indicator $x_i$ through the remaining $n-1$ indicators. | $T_i$ | the first-order sensitivity index |

**Table 3.** *Cont.*

| Variable | Interpretation | Variable | Interpretation |
|---|---|---|---|
| $T_{ijk\,n}$ | the coupling effect of indicator $x_i$, along with other indicators | $W_j$ | the weight value for the indicator combination |
| $W_j'$ | he subjective weight result produced from AHP | $W_j''$ | the objective weight result derived from the EFAST algorithm. |
| $r_{ij}$ | the $j$th evaluation index's standardized data from the $i$th program | $a_{ij}$ | denotes the index's original data from the same program |
| $J_1$ | the collection of data based on benefits | $J_2$ | the collection of indicators based on costs. |
| $S_i$ | the group benefit value | $R_i$ | the individual regret value |
| $d(\tilde{z}_j^+, r_{ij})$ | the index distance | $d(\tilde{z}_j^+, \tilde{z}_j^-)$ | the standard distance |
| $v$ | the coefficient of the maximum group utility decision strategy | $R_j$ | the categorization indication's weight for the $i$th indicator. |
| $W_i$ | The combined indicator's weight of it. | $V_{p*}$ | the relative preference relation operator |
| $Q_i$ | stands for water cycle health | $P_j$ | the indicator's barrier degree |

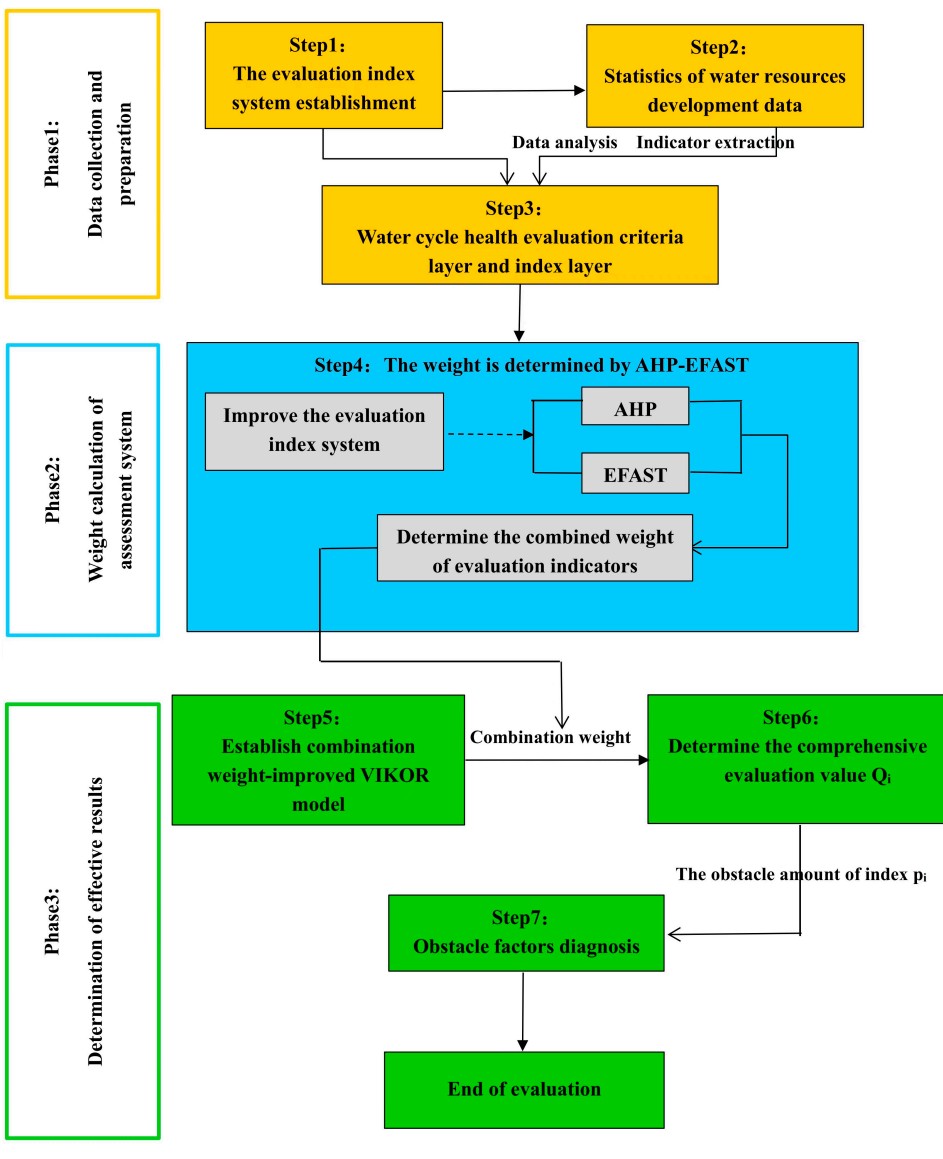

**Figure 4.** Flow chart for the assessment of water cycle health.

## 3. Results and Discussion

### 3.1. Weight Calculation Results

The objective and subjective weights of each index were determined using the EFAST algorithm and the AHP algorithm, respectively. Then, the minimum information entropy principle was used to combine the subjective and objective weights, and the final result was the comprehensive weight values for each index. Figure 5 displays the total weight values for the 22 assessment system indications.

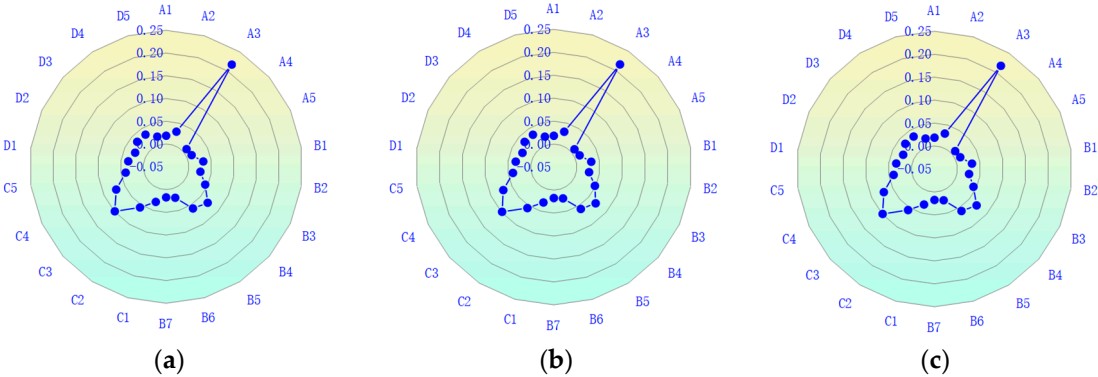

**Figure 5.** Zheng-Bian-Luo region combined weight radar map. (**a**) Zhengzhou (**b**) Kaifeng (**c**) Luoyang.

### 3.2. Evaluation of Index Layer

Each indicator of the health evaluation of the water cycle in the Zheng-Bian-Luo area from 2011 to 2021 is dimensionless in terms of being processed, and a health analysis of the indicators was carried out for each indicator after processing based on the specified zoning criteria. The health situation of each indicator in the Zheng-Bian-Luo area is derived year by year, and the graphical representation of the water cycle indicators in the Zheng-Bian-Luo area is, thus, derived.

The health status of each indicator is calculated based on the indication thresholds. Figure 6 displays the indicator layer's state of health. According to the indicator layer evaluation results, the Zheng-Bian-Luo region is primarily arid and semi-arid; annual precipitation (A3) is unstable with significant regional variations; annual precipitation in Kaifeng gradually increases; annual precipitation in Luoyang tends to be stable, but annual precipitation in Zhengzhou is extremely unstable. Additionally, the total water resources (A2) in the Zheng-Bian-Luo area are mostly in a general state due to the significant differences in the local water production system (A4), but the rate of the living drinking water standard (C2) and water function area standard (C3) are developing into a reasonable trend, which shows that the Zheng-Bian-Luo area places a high priority on the protection of water resources and has a comprehensive water resource management system. Additionally, the Zheng-Bian-Luo area's per capita comprehensive water consumption (B3) and the degree of health of agricultural water consumption (A5) are both improving year after year. In order to lessen the negative impact on water circulation caused by the development in urbanization, China is now increasing the coverage of green space in built-up regions (D1) and enhancing the storage capacity of rivers and lakes (D3). In order to lessen the negative effects of human growth on the environment, the Zheng-Bian-Luo region is progressively adjusting the sewage treatment rate (C5) and the density of drainage pipes in built-up areas (D4).

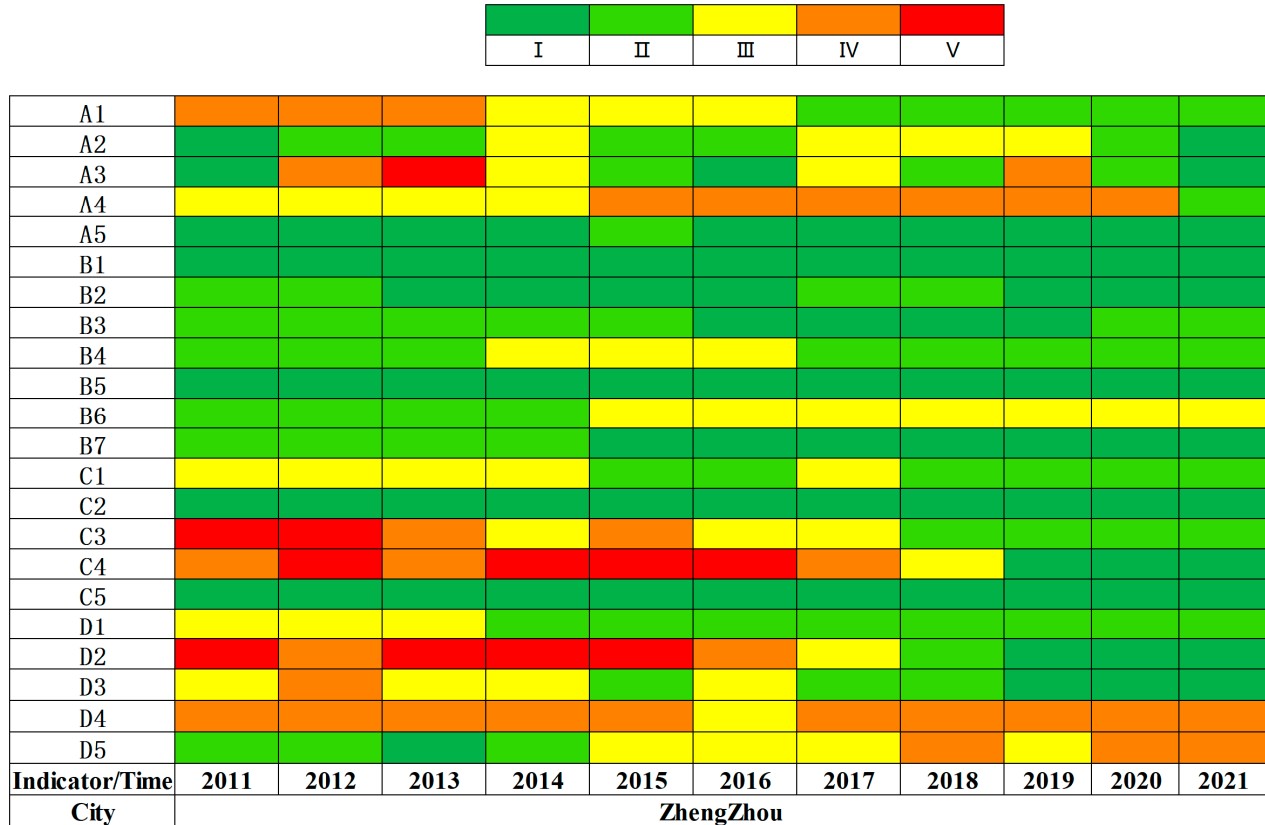

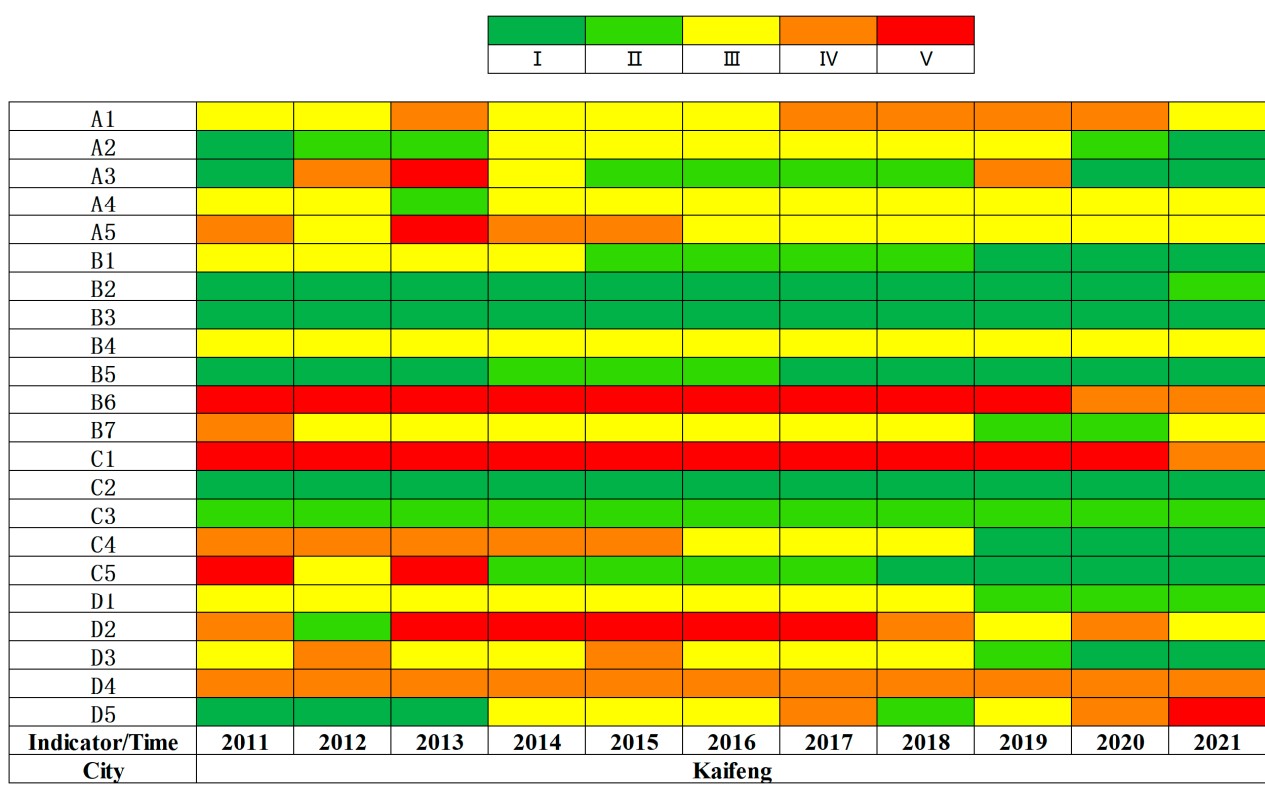

**Figure 6.** *Cont.*

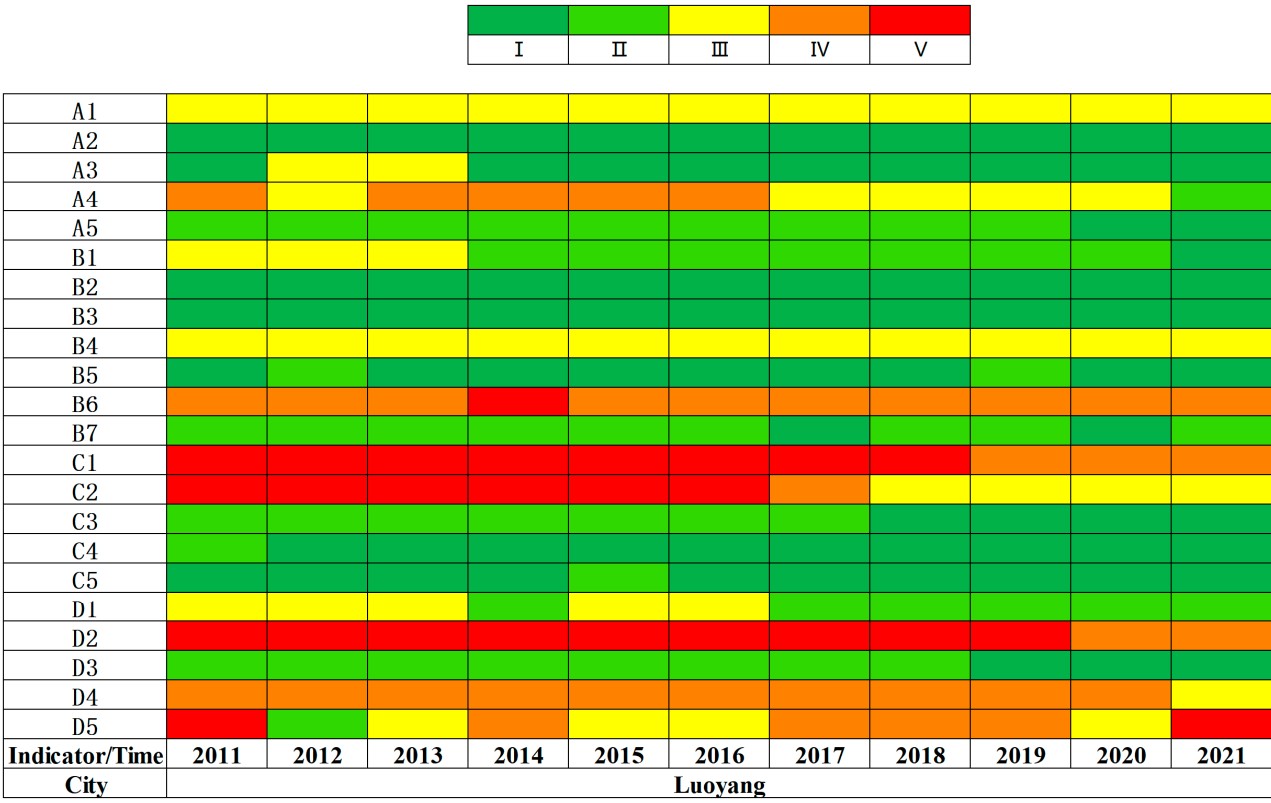

**Figure 6.** Schema showing the outcomes of each index's evaluation.

Corresponding to the health trend is the proportion of river length above class III water quality (C4), and the health degree of C4 in the Zheng-Bian-Luo area is gradually developing towards a healthy trend. The discharge of human sewage (C1) has caused a certain degree of pollution to water quality. However, due to the government's introduction of some water resources protection policies, the Zheng-Bian-Luo area (C1) has gradually developed towards a better trend.

The variation in shallow groundwater level (D5) serves as a proxy for significant changes in health status. In between the five stages of health, sub-health, general, sub-morbidity, and morbid state, Zhengzhou and Luoyang alternately fluctuate. The Kaifeng region is slowly growing worse and has reached a dismal state. However, the proportion of groundwater supply (A1), and its health status is mostly general and sub-morbid, indicating that although the D5 of each region fluctuates greatly, the impact on A1 is not obvious at present. However, we can not only rest on the status quo but also need to take precautions. Therefore, we still need to improve. The specific improvement measures include appropriately reducing the exploitation of groundwater and adjusting the industrial structure. The average water use per mu of irrigated farmland (B2) (1 mu = 666.67 square meters), the utilization coefficient of irrigation water (B4), the water consumption per CNY 10,000 of GDP (B7), and the water consumption of the ecological environment (D2) in the Zheng-Bian-Luo area are all moving in the direction of a reasonable trend, indicating that the water use structure in Zheng-Bian-Luo is reasonable. The majority of the water penetration rate (B5) in the Zheng-Bian-Luo area is in a healthy or sub-healthy state, which is a clear indication that the locals are very conscious of water conservation and water protection.

### 3.3. Evaluation of Target Layer

3.3.1. The Calculation Result of the Comprehensive Evaluation Value $Q_i$

The improved VIKOR model used in this paper was used to calculate the comprehensive evaluation value, $Q_i$, of the Zheng-Bian-Luo area and was then used to evaluate the

health of the water cycle in the Zheng-Bian-Luo area. Table 4 presents the outcomes. The water cycle health level is divided into five levels, and its four critical values are given. The health level of the water cycle in the Zheng-Bian-Luo area is shown in Table 5.

**Table 4.** Comprehensive assessment value calculation findings for the Zheng-Bian-Luo area.

| City/Years | 2011 | 2012 | 2013 | 2014 | 2015 | 2016 | 2017 | 2018 | 2019 | 2020 | 2021 |
|---|---|---|---|---|---|---|---|---|---|---|---|
| Zhengzhou | 0.464 | 0.871 | 0.863 | 0.697 | 0.689 | 0.389 | 0.474 | 0.340 | 0.289 | 0.284 | 0.000 |
| Kaifeng | 0.449 | 0.821 | 0.965 | 0.874 | 0.753 | 0.673 | 0.514 | 0.534 | 0.373 | 0.227 | 0.000 |
| Luoyang | 0.428 | 1.000 | 0.674 | 0.444 | 0.627 | 0.609 | 0.390 | 0.263 | 0.312 | 0.270 | 0.000 |
| Average | 0.447 | 0.897 | 0.834 | 0.672 | 0.690 | 0.557 | 0.459 | 0.379 | 0.325 | 0.260 | 0.000 |
| Level I Critical value | 0.18 | 0.17 | 0.18 | 0.17 | 0.16 | 0.16 | 0.19 | 0.15 | 0.15 | 0.14 | 0.14 |
| Level II Critical value | 0.26 | 0.27 | 0.25 | 0.29 | 0.28 | 0.29 | 0.29 | 0.28 | 0.28 | 0.29 | 0.29 |
| Level III Critical value | 0.42 | 0.43 | 0.42 | 0.41 | 0.44 | 0.45 | 0.44 | 0.42 | 0.45 | 0.44 | 0.44 |
| Level IV Critical value | 0.85 | 0.83 | 0.84 | 0.85 | 0.79 | 0.78 | 0.79 | 0.81 | 0.77 | 0.78 | 0.78 |

**Table 5.** Water cycle health level evaluation results for the Zheng-Bian-Luo area.

| City/Years | 2011 | 2012 | 2013 | 2014 | 2015 | 2016 | 2017 | 2018 | 2019 | 2020 | 2021 |
|---|---|---|---|---|---|---|---|---|---|---|---|
| Zhengzhou | IV | V | V | IV | IV | III | IV | III | III | II | I |
| Kaifeng | IV | IV | V | V | IV | IV | IV | IV | III | II | I |
| Luoyang | IV | V | IV | IV | IV | IV | III | II | III | II | I |

3.3.2. Analysis of the Findings of the Water Cycle Health Assessment

Tables 3 and 4 show that from 2011 to 2021, the Zheng-Bian-Luo region's water cycle has progressed in a positive direction. The average value of the comprehensive evaluation result $Q_i$ (the distance from the ideal solution) has shortened from 0.447 to 0.00 (the distance from the ideal solution is 0, meaning the ideal solution is 0). The overall situation is improving, but there are still variances in the degree of the water cycle's health in the Zheng-Bian-Luo region. The comprehensive evaluation results of the Zheng-Bian-Luo area are shown in Figure 7.

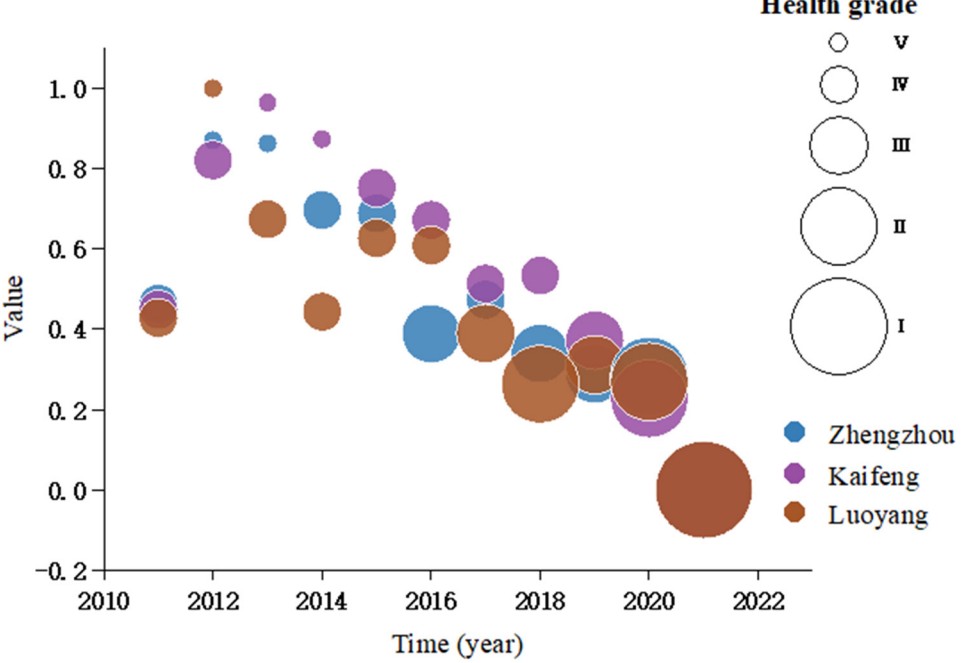

**Figure 7.** A bubble diagram of the full evaluation results for the Zheng-Bian-Luo region.

In the time dimension, the water cycle level in the Zheng-Bian-Luo region decreased from 2011 to 2015, with a major recovery from 2015 to 2021. There is no obvious difference in the rate of improvement in the Zheng-Bian-Luo area. The water cycle health level of the Zheng-Bian-Luo area has gradually improved from the initial sub-pathological state to a healthy state.

In terms of the spatial dimension, the water cycle health level of the Zheng-Bian-Luo area is better, and the three cities reached a healthy (grade I) level by 2021, and the spatial difference is not significant. Water cycle health in the Zheng-Bian-Luo region was sub-morbid in 2011. In 2013, the health of the water cycle in the Zheng-Bian-Luo area became more serious. The three cities' water health has progressively improved since 2016, but the impact is not yet readily apparent. The three cities' water cycle health has greatly improved in recent years (from 2019 to 2021). The spatial and temporal distribution map of water cycle health in the Zheng-Bian-Luo area can be seen in Figure 8.

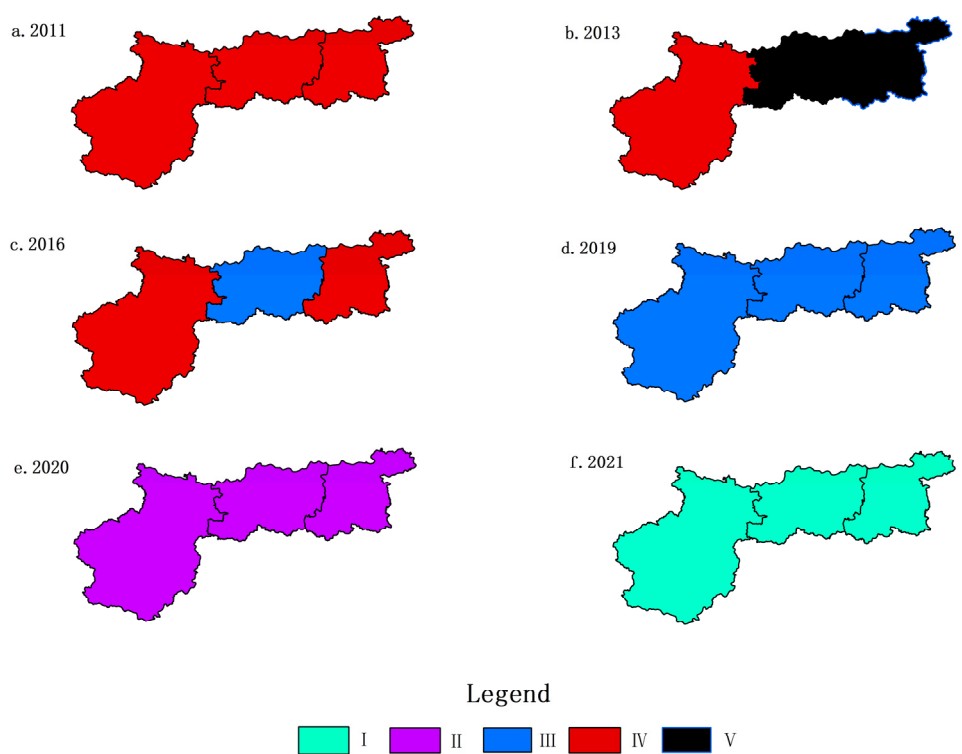

**Figure 8.** Spatial and temporal distribution map of Zheng-Bian-Luo's water cycle health.

The outcomes of the index layer were further assessed based on the temporal dimension and the geographical dimension. In the time dimension, the average water consumption per mu of farmland irrigation, the per capita comprehensive water consumption, the irrigation water utilization coefficient, the water penetration rate, the river and lake storage capacity, and other indicators from 2011 to 2021 are relatively stable. The proportion of agricultural water, the water consumption of CNY 10,000 industrial-added value, the compliance rate of water functional areas, the proportion of river length above class III water quality, the coverage rate of green space in built-up areas, the storage capacity of rivers and lakes, the compliance rate of drinking water sources, the proportion of groundwater supply, and other indicators have significantly improved and developed towards a good trend. This demonstrates how human activity is gradually lessening its burden on the water cycle's health in the temporal dimension. Environmental indicators, such as environmental status, are trending in a positive direction as more individuals safeguard the environment. Zhengzhou, Kaifeng, and Luoyang's water cycle health is improving in the geographical dimension, but the three cities' development priorities are

distinct. While Zhengzhou's economy is growing quickly, and there is a greater need for water, Kaifeng and Luoyang have stronger endowments in terms of water resources. As a result, there are still some regional disparities in the water cycle's overall health in the Zheng-Bian-Luo area.

*3.4. Factor Analysis of Water Circulation Health Disorders*

In order to more clearly illustrate the size distribution of the obstacle degree of the indexes, the obstacle degree of each index in the Zheng-Bian-Luo area from 2021 was calculated using Formulas (24)–(26), and the eight indexes with the highest obstacle degree were listed as the main obstacle factors in Table 6. Table 6 shows that there are bigger obstacle degrees with regard to the drinking water source compliance rate, the per-person comprehensive water consumption, the water consumption per CNY 10,000 of industrial-added value, the water consumption per ten CNY 10,000 of GDP, the average water consumption per mu of farmland irrigation, the domestic water consumption rate, the proportion of groundwater water supply, and the proportion of agricultural water in the Zheng-Bian-Luo area.

**Table 6.** Obstacle degree of main obstacle factors of water resource carrying capacity in the Zheng-Bian-Luo area in 2021.

| City/Indicator | C2 | B3 | B1 | B7 | B2 | B6 | A1 | A5 |
|---|---|---|---|---|---|---|---|---|
| Zhengzhou | 32.38 | 15.94 | 11.65 | 4.48 | 3.42 | 1.43 | 1.05 | 0.67 |
| Kaifeng | 32.83 | 4.69 | 11.81 | 5.19 | 9.47 | 7.62 | 0.64 | 0.62 |
| Luoyang | 0.42 | 31.96 | 23.35 | 9.36 | 16.05 | 15.07 | 1.97 | 1.29 |

Figure 9 depicts the link between the primary barriers in the Zheng-Bian-Luo region and the water cycle's overall health. In terms of the first obstacle factor in the Zheng-Bian-Luo area, Luoyang encounters the per capita comprehensive water consumption first, and the rest of the area encounters the standard rate of drinking water source first. In terms of the second obstacle factor, Zhengzhou sees per capita comprehensive water consumption, and the remaining areas encounter water consumption of CNY 10,000 of industrial-added value. In terms of the smallest obstacle factor, Luoyang sees the standard rate of drinking water sources, and the remaining two cities see the proportion of agricultural water. The reasons for this are that Luoyang is rich in local water resources, there are more mountainous areas in the area, and the water resources are less polluted.

The following focused recommendations are made in conjunction with the analytical findings of the water cycle health degree and obstacle factors in the Zheng-Bian-Luo area. The objective is to enable the Zheng-Bian-Luo area to quickly achieve regional sustainable development and to enhance the existing state of water cycle health in the area.

(1) The main influencing factor of Zhengzhou and Kaifeng is the compliance rate of drinking water (C4), which also confirms that the two cities are currently using the water source provided by the "South-to-North Water Diversion" project. Therefore, the key to improving the health of the Zhengzhou and Kaifeng water cycle is to increase the protection of the ecological environment, increase the restoration of the wetlands, and block the various river sewage outlets to reduce water pollution. Comprehensive water use per person (B3) and the water use of CNY 10,000 industrial-added value (B1) are the two primary variables affecting Luoyang. As a result, enhancing the development and usage of water resources and distributing water sensibly is essential in improving the health of the water cycle in Luoyang.

(2) For cities with rapid economic development, we should strengthen the ecological protection of such cities, continue to promote water-saving equipment and enhance citizens' awareness of water-saving. Cities with good water resource conditions should give full play to their natural advantages and develop and utilize water resources scientifically and rationally according to the actual local situation so as

to achieve the purpose of sustainable development. In the process of economic development, underdeveloped areas should pay attention to the protection of the ecological environment and achieve harmony between man and nature.

(3) All localities should strengthen communication and exchanges. Cities with good water cycles should actively promote their own experience, while cities with poor water cycles should combine their own reality, improve, accordingly, on this basis, and adjust the industrial structure appropriately so as to improve their water cycle and promote sustainable development in the region.

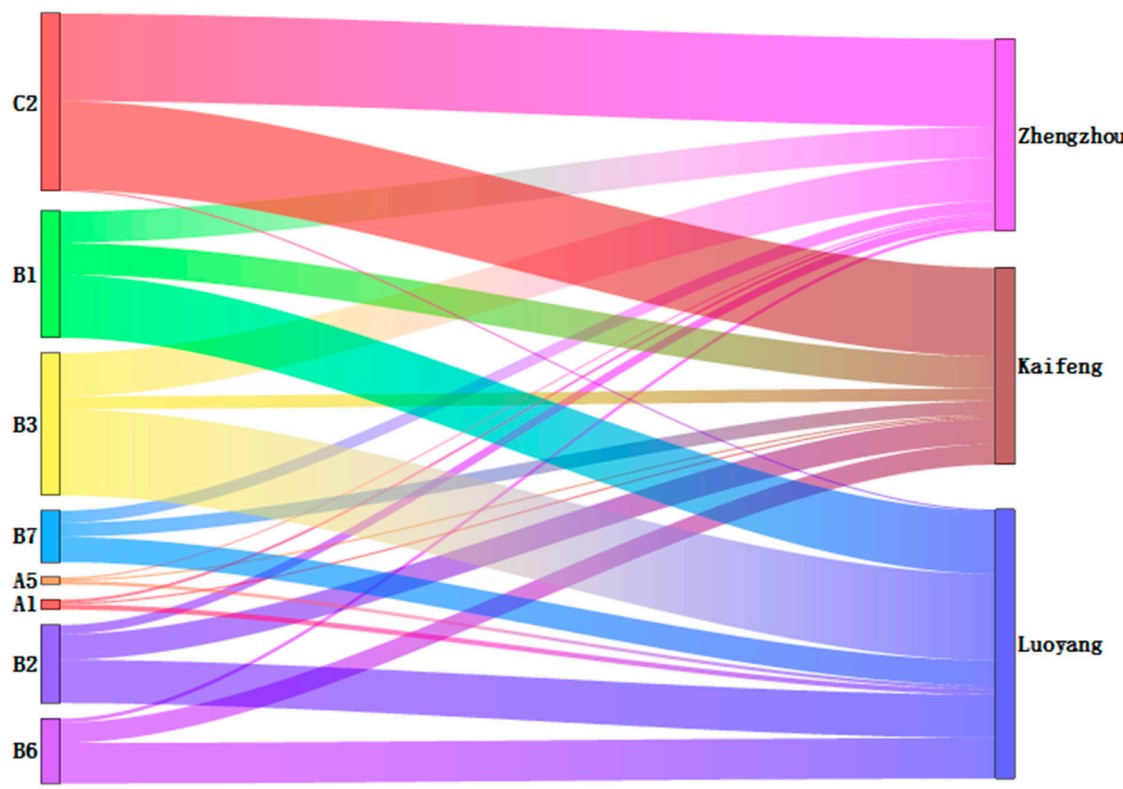

**Figure 9.** Analysis of the Zheng-Bian-Luo region water cycle hindrance factors in 2021.

*3.5. Method Validity Test*

In this study, the findings from the water cycle health assessment conducted in the Zheng-Bian-Luo region between 2020 and 2021 were chosen, and the results produced by the FCE technique and the conventional VIKOR model were compared to those obtained by the upgraded VIKOR model. Its goal is to evaluate the revised VIKOR model's applicability, stability, and dependability. The outcomes are displayed in Table 7:

**Table 7.** Evaluation findings comparison.

| Evaluation results of three cities of Zheng, Bian, and Luo in 2020. | | | |
|---|---|---|---|
| **City** | **FCE** | **VIKOR** | **Improved VIKOR** |
| Zhengzhou | II | III | II |
| Kaifeng | II | II | II |
| Luoyang | II | III | II |
| Evaluation results of three cities of Zheng, Bian, and Luo in 2021. | | | |
| **City** | **FCE** | **VIKOR** | **Improved VIKOR** |
| Zhengzhou | I | I | I |
| Kaifeng | II | I | I |
| Luoyang | I | I | I |

Table 7 shows that the results achieved by the three approaches are quite similar. When studying water cycle health in the Zheng-Bian-Luo area in 2020, there are still some disparities between the conventional VIKOR model and the upgraded VIKOR model. The relative preference relationship was used in the improved VIKOR model in place of the $L_p - metric$ aggregation function and the distance between the positive and negative ideal solutions were taken into account. This avoids the traditional VIKOR model's use of the positive ideal solution alone in the actual evaluation process, which results in falsely high actual evaluation values and an inability to accurately evaluate the data of each index. The FCE technique has a high degree of subjectivity and is dependent on subjective approaches, including expert personal views, questionnaires, and on-site grading. The revised VIKOR model, on the other hand, is based on facts and can correctly and objectively depict the benefits and drawbacks of any scheme [30,31]. It is clear from an analysis of the healthy water cycle in the Zheng-Bian-Luo area in 2021 that the upgraded VIKOR model is more stable than the conventional model. The updated VIKOR model has higher application and stability in the process of evaluating the health of the water cycle thanks to the aforementioned study.

## 4. Conclusions

The definition and assessment of a healthy water cycle can increase human comprehension of the water cycle and influence societal development in the right direction. In this paper, with the help of the "natural attributes" and "social attributes" of the water cycle, the four criterion layers of water abundance (A), water utility (B), water quality (C), and water ecology (D) were taken as the entry points, and 22 indicators were selected according to the Zheng-Bian-Luo area to construct a water cycle health evaluation system. The water cycle health of the Zheng-Bian-Luo region was examined from 2011 to 2021 using the updated VIKOR model, FCE, and original VIKOR model for comparison, verification, and obstacle factor analysis. The findings of this investigation are as follows:

(1) Between 2011 and 2021, the Zheng-Bian-Luo region's water cycle health status was expected to increase significantly. In the Zheng-Bian-Luo region, the average comprehensive evaluation value, $Q_i$ (distance from ideal solution), decreased from 0.447 in 2011 to 0.00 in 2021 (distance from the perfect solution is 0, which is the ideal solution). There are some differences in the health level and improvement rate of the water cycle between different regions, but the overall comprehensive evaluation value has been improved after 2014, indicating that all regions have adopted a series of measures suitable for local development according to their own actual situation, reflecting the observation of the trend of indicators. It is useful for the choice of future planning actions and also reflects the importance of human beings to the healthy development of the water cycle.

(2) The evaluation results of the health of the water cycle in the Zheng-Bian-Luo region show high consistency across the three techniques. However, each year's health status varies a little bit. The $L_p - metric$ aggregation function was replaced by a relative preference relationship in the improved VIKOR model, and the distance between the positive and negative ideal solutions was taken into account. As a result, the traditional VIKOR model's use of only the positive ideal solution, which results in falsely high actual evaluation values, cannot objectively and steadily evaluate the data of each index; this was avoided by using the improved VIKOR model.

(3) The compliance rate of drinking water sources, the per capita comprehensive water consumption, the water consumption per CNY 10,000 of industrial-added value, the water consumption per CNY 10,000 of GDP, the average water consumption per mu of farmland irrigation, the domestic water consumption rate, the proportion of groundwater supply, and the proportion of agricultural water are the main factors obstructing the improvement of water cycle health in the Zheng-Bian-Luo area. The key to raising the water cycle's health level in the Zheng-Bian-Luo region will be to handle this set of indications with science and reason.

**Author Contributions:** Conceptualization, M.Z. and J.W.; methodology, Y.H.; software, J.L.; validation, J.W., J.L. and M.Z.; formal analysis, J.W.; investigation, Y.H.; resources, M.Z.; data curation, J.W.; writing—original draft preparation, J.L.; writing—review and editing, M.Z.; visualization, J.L.; supervision, J.W.; project administration, M.Z.; funding acquisition, M.Z. All authors have read and agreed to the published version of the manuscript.

**Funding:** This research was funded by the National Natural Science Foundation of China (52009045), the scientific and technological research projects in Henan Province (212102310539), and postdoctoral funding in Henan Province (202102086).

**Data Availability Statement:** Not applicable.

**Conflicts of Interest:** The authors declare no conflict of interest.

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
