# Peer review of "Water Cycle Health Assessment Using the Combined Weights and Relative Preference Relationship VIKOR Model: A Case Study in the Zheng-Bian-Luo Region, Henan Province"

_water, doi:10.3390/w15122266_

Round 1
Reviewer 1 Report
Dear Authors
I congratulate you on the research proposal submitted. The topic studied was the improvement of water quality in Zheng-Bian-Luo region. The authors developed a model that used the AHP method to measure the weights of criteria combined with the VIKOR method. The authors compared the model with the VIKOR method. The studied topic is relevant and I believe that it contributes to the understanding of the process of water quality improvement in Zheng-Bian-Luo region, I believe that it can be a model to be exported. However, I observed some points that can be improved, which I transcribe below:
1. In the introduction the authors approached concepts about the subject studied, and made an approach to the multicriteria methods they used in the development of the model. Some issues need to be improved:
1.1 - I did not observe the statement of the problem question that leads the research. For the reader, the problem is not clear. Likewise, the objectives were not stated. It is suggested that the problem that motivated the article and the objectives of the work be explicitly described in the introduction;
1.2 The authors chose not to use a section dedicated to the literature review. 1.2 The authors chose not to use a section dedicated to the literature review, and to compensate for this, they wrote a summarized review in the introduction. This review is very poor. I suggest that they should deepen the researches published in the last 5 years on the subject. In relation to the multicriteria methods, the authors only cited a few researches related to the theme. I suggest that they expand it, indicating other methods and their areas of action. I suggest the reading of a research published in 2022 that made a broad review of several multicriteria methods, I believe it will serve to complement this issue ("A Systematic Review of the Applications of Multi-Criteria Decision Aid Methods (1977-2022)");
1.2.1 Regarding the proposed model, the authors must insert a brief paragraph in which they justify the choice of EFAST, AHP and VIKOR methods. Why not choose other methods to obtain the weights of criteria, such as CILOS, IDOCRIW, FUCOM, LBWA, SAPEVO-M, and MEREC. In this sense, I suggest reading the following paper: ("A Comprehensive Review of the Novel Weighting Methods for Multi-Criteria Decision-Making");
1.3 I suggest inserting at the end of the introduction a short paragraph that summarizes the other sections that compose the paper.
2. How were the criteria chosen? Were the criteria chosen based on research? I suggest you insert this clarification in the text.
3. I suggest that the authors insert a list of abbreviations and variables used.
I wish you a good review.
Author Response
Response to Reviewer 1 Comments
Thank you very much for taking the time to revise our paper. We have given all the answers to your questions.The contents are as follows :
Point 1: I did not observe the statement of the problem question that leads the research. For the reader, the problem is not clear. Likewise, the objectives were not stated. It is suggested that the problem that motivated the article and the objectives of the work be explicitly described in the introduction.
Response 1:Thank you for taking time to revise our paper.In the introduction, we have added clear questions and work objectives that cause the article.The contents are as follows :There are few studies on the comprehensive evaluation of water cycle health in urban agglomerations, and few studies analyze the water cycle health of urban agglomerations from the perspective of time and space.A healthy water circulation system is the premise and foundation for the high-quality development of urban agglomerations, and the evaluation of water circulation health is the key link to improve the health of urban agglomerations.Therefore, the evaluation of the current water resources situation and urban development model of urban agglomerations can provide a theoretical basis for the sustainable utilization of water resources and regional sustainable development of urban agglomerations.
Point 2:The authors chose not to use a section dedicated to the literature review. 1.2 The authors chose not to use a section dedicated to the literature review, and to compensate for this, they wrote a summarized review in the introduction. This review is very poor. I suggest that they should deepen the researches published in the last 5 years on the subject. In relation to the multicriteria methods, the authors only cited a few researches related to the theme. I suggest that they expand it, indicating other methods and their areas of action. I suggest the reading of a research published in 2022 that made a broad review of several multicriteria methods, I believe it will serve to complement this issue ("A Systematic Review of the Applications of Multi-Criteria Decision Aid Methods (1977-2022)");
Response 2:Thank you for taking time to revise our paper.We have made changes in accordance with your views.We specially marked the modified place yellow so that you can easily see.The contents are as follows:’Numerous domestic specialists and academics started researching the health of the urban water cycle based on the general trend of water resources, water ecology, and water environment as the problem of water resources became more and more apparent. Zhang J et al. [4] first proposed the concept of a healthy water cycle, mainly emphasizing the use of recycled water and the popularization of wastewater purification as the key to a healthy water cycle. Xu Xiangjun et al. [5] examined the idea of an urban water cycle and suggested ways to build one in accordance with the elements that influence it. Some researchers have used the binary characteristics of water resources to build an assessment index system. For example, Tang Jizhang et al. [6] studied the health of the water cycle in Xi'an based on the principle of "target criteria indicator", which is based on the attributes of the binary water cycle in the city.The results show that improving the rationality of water resources development and utilization and alleviating the contradiction between supply and demand of water resources are the key to improve the health of water cycle in Xi 'an. Some experts and scholars evaluate the water cycle health of cities or regions based on the coupling of the natural and social aspects of the water cycle. Wang [7] evaluated the health of the water cycle of farmland from four dimensions: water source, water extraction and transmission subprocess, water consumption subprocess, and drainage retreat subprocess.The results show that the health degree of farmland in the irrigation area is developing towards a good trend after a series of measures have been taken by human beings.Chen Jiongli [8] gave an evaluation of the water cycle in Yinchuan City from four dimensions, including the water ecology level, water environment quality, water resource abundance, and water resource utilization.The results show that the differences in each dimension are small and are developing towards a good trend, and it is concluded that the amount of sewage regeneration is the main index affecting the health of water cycle in Yinchuan.Ma Jing et al. [9] evaluated the water cycle from five perspectives: water source, water supply, water use, drainage, and reuse. The results show that although each dimension has some fluctuations, the overall trend is improving. It also shows that the adjustment of industrial structure and the implementation of water conservancy and people 's livelihood policies in Handan City are important factors affecting the health of water cycle in Handan City.Other specialists and academics have examined the health of the water cycle using a variety of evaluation techniques and views based on their own expertise. Yang Haiyan et al. [10] evaluated the water carrying capacity of Weifang City based on the VIKOR method, and the results showed that the water carrying capacity of some areas in Weifang City did not match the local water conditions. Li Yinjiu et al. [11] evaluated the health degree of the Guangdong River based on a composite fuzzy matter element VIKOR model. The study showed that the quality of the river water was the main factor affecting the health level of the river. Li Na et al. [12] evaluated the sustainability of packaging schemes based on the entropy weight-VIKOR model. The results show that the packaging scheme with paperboard as the main material has the best sustainability.He Gang et al. [13] evaluated the water and soil ecological security of mining cities in Anhui Province based on the VIKOR model. The results show that the results obtained by VIKOR are in good agreement with the actual results, and the per capita water consumption is the main factor affecting the water and soil ecological security of mining cities in Anhui Province.Wang Lunyan et al. [14] evaluated the water resources carrying capacity of nine provinces and regions in the Yellow River Basin based on a fuzzy set-pair analysis method with combined weights and a barrier degree model. The study showed that the overall improvement trend of the carrying capacity of the Yellow River Basin provinces was obvious. Bai Fangfang et al. [15] objectively evaluated the utilization efficiency of agricultural water resources in nine provinces and regions in the Yellow River Basin based on the entropy weight TOPSIS model. The study showed that the overall agricultural water resource utilization efficiency of each province improved, and the gap in agricultural water resource utilization efficiency between the provinces became significantly smaller. Despite the fact that the water cycle has been researched in other nations in the past, there has not been as much research carried out regarding its health, and only a few academics have looked into the best ways to assess this. Gani et al. [16] used distance estimation to govern the urban water cycle. Deng et al. [17] gave an evaluation of the health of the Taihu Lake Basin in China based on an improved entropic fuzzy material element model.Meneses et al. [ 18 ] evaluated the urban water cycle based on the life cycle assessment method, revealing that the non-drinking water use of reclaimed water in the urban water cycle has environmental and economic advantages.Pinto et al. [19] evaluated river health based on factor analysis. According to the study, eutrophication, microbiological contamination, and anaerobic fermentation are the key issues that have an impact on the health of rivers.
Point 3: Regarding the proposed model, the authors must insert a brief paragraph in which they justify the choice of EFAST, AHP and VIKOR methods. Why not choose other methods to obtain the weights of criteria, such as CILOS, IDOCRIW, FUCOM, LBWA, SAPEVO-M, and MEREC. In this sense, I suggest reading the following paper: ("A Comprehensive Review of the Novel Weighting Methods for Multi-Criteria Decision-Making")
Response 3:The essence of subjective weighting methods such as AHP and ANP is that decision makers subjectively determine the weight of each index based on experience. Although its explanatory is strong, its objectivity is poor.The essence of objective assignment methods such as entropy weight method and EFAST method is that the original data of calculating weights are obtained from the actual data of evaluation indicators in the process of evaluation. Although the objective assignment method to determine the weight accuracy is higher, but it is contrary to the actual situation, and poor interpretation.Therefore, subjective and objective comprehensive weights are adopted. That is, subjective assignment method AHP and objective assignment method EFAST.According to the valuable literature provided by you, we believe that the methods of CILOS, IDOCRIW, FUCOM, LBWA, SAPEVO-M and MEREC are also reasonable, but the methods of determining weights are representative.In future research, we will take more valuable methods that you provide in order to make our research more perfect and reasonable.
Point 4: suggest inserting at the end of the introduction a short paragraph that summarizes the other sections that compose the paper.
Response 4:
Introduction Part:Introduce the background of the research problem, the problem and the selection of methods.
Materials and Methods Part:This paper introduces the detailed information of Zheng-bian-luo urban agglomeration in the study area, as well as the selection of indicators, the search of data and the specific operation of the method.
Results and Discussion Part:The results calculated by the method are analyzed from three perspectives : target layer, dimension layer and index layer, and then the obstacle degree model is used for analysis, and finally compared with other methods.
Conclusions Part:Based on the above analysis, draw conclusions.
Point 5:. How were the criteria chosen? Were the criteria chosen based on research? I suggest you insert this clarification in the text.
Response 5:In the evaluation system, the selection of index layer mainly follows two standards.Firstly, according to the existing research results, the indicators with high correlation with water cycle health are selected, which can make the selected indicators scientific.Second, according to the characteristics of the study area, select regional representative indicators, so that the selected indicators are independent and representative.
Point 6: I suggest that the authors insert a list of abbreviations and variables used.
Response 6:Thank you for taking time to revise our paper.We have made changes to your valuable comments.The contents are as follows.
|
Variable |
Interpretation |
Variable |
Interpretation |
|
xij |
the evaluation index's initial value |
rij |
the index's value |
|
X(—)j |
the jth index's mean value |
Sj |
its standard deviation. |
|
V |
the overall variance of the model's output |
the variance of the indicator |
|
|
the variance of the interaction between the indicators and |
the variance of the interaction between the indicators , , and |
||
|
the variance of the interaction of the indicator through the remaining n-1 indicators.
|
the first-order sensitivity index |
||
|
|
the coupling effect of indicator , along with other indicators |
the weight value for the indicator combination |
|
|
he subjective weight result produced from AHP |
the objective weight result derived from the EFAST algorithm.
|
||
|
the jth evaluation index's standardized data from the ith program |
denotes the index's original data from the same program |
||
|
the collection of data based on benefits |
the collection of indicators based on costs.
|
||
|
the group benefit value |
the individual regret value |
||
|
the index distance |
the standard distance |
||
|
the coefficient of the maximum group utility decision strategy |
the categorization indication's weight for the ith indicator. |
||
|
The combined indicator's weight of it.
|
the relative preference relation operator |
||
|
stands for water cycle health |
the indicator's barrier degree |

Reviewer 2 Report
see file

Author Response
Response to Reviewer 2 Comments
Thank you very much for taking the time to revise our paper. We have given all the answers to your questions.The contents are as follows :
Main comments:
Point 1:I am not an expert in multi-criteria modeling,so I assume that all the technical passages present in the paper are correct and I consider this part very important for the positive evaluation of the entire paper.My main remark concerns what is reported in lines 328-329,in which the variation over time of indicator D5,relating to the level of the surface water,is commented as a modesr change in health starus'.In effect,it is a trend that leads 2 out of 3 sites to switch from I (healthy)to V(pathological) The very object of the indicator (the level of the groundwater)is generally considered particularly significant in any analysis of the environmental state that one wishes to pursue.It is important that this point is clarified.
Response 1:Thank you very much for taking the time to modify our paper, we have made changes to your questions, as follows :The variation in shallow groundwater level (D5) serves as a proxy for significant changes in health status. In between the five stages of health, subhealth, general, submorbidity, and morbid state, Zhengzhou and Luoyang alternately fluctuate. The Kaifeng region is slowly growing worse and has reached a dismal state. However, the proportion of groundwater supply ( A1 ), its health status is mostly general and sub-morbid, indicating that although the D5 of each region fluctuates greatly, the impact on A1 is not obvious at present. However, we can not only rest on the status quo, but also need to take precautions. Therefore, we still need to improve. The specific improvement measures include appropriately reducing the exploitation of groundwater and adjusting the industrial structure.
Point 2:Furthermore,it is not clear to me the link between the results shown,the actions taken by the authorities to achieve them and people's awareness,repeatedly reported (e.g.lines 318-319,320-321,369-370,384-385,506-507).These citations are of no use for the purpose of transferring the experience in question to other different locations,as the paper concerns only the evaluation of the evolution over time of the state of the system,but not of the measures put in place to achieve of these goals.So,it might be enough to mention once the fact that good results depend on good deeds of people and authorities.It is important that in a scientific writing knowledge is shared and opinions are not reported that are not supported by objective and repeatable data.
Response 2:The results shown in the article, on the one hand, compared with other methods, prove the universality of the method. On the other hand, through the data, it points out that when the Zheng-Bian-Luo urban agglomeration is developing, in what aspects it is doing well, it will continue to maintain in the future. For those who do not do well, it can attract people 's attention in the future development and take a series of necessary measures to improve. It plays a guiding role in how people will develop towards a healthy trend when building cities and developing the economy in the future.
Detailed comments:
Point 1:In several parts of the paper,starting from the summary (line 31 cfindings),a spurious c appears attached to the word (other examples at lines 38,39,40).Please,check carefully.
Response 1:Thank you very much for taking the time to revise our paper. We have revised your questions and carefully examined them.
Point 2:In several parts of the paper a unit of measurement of areas,mu,is reported (e.g.at line336).Areas should be in square m,in are or in hectare (ha),or you must first indicate the value of mu (e.g.’it corresponds to 1/15 of a hectare,or about 666.67m2’)
Response 2:The area measurement unit mentioned in this paper is the average water consumption per mu of farmland irrigation ( B2 ), but its unit is a volume of measured water consumption, that is, cubic meters, not an area unit. We have clearly marked in the text : 1 mu = 666.67 square meters.
Point 3:Line 374:is reported the expression ‘from submorbidity to morbidity’.Please,make it coherent with the rest of the paper.
Response 3:Thank you very much for taking the time to revise our paper.We have modified the expression of line 374, which is consistent with the expression of other parts of the paper.

Round 2
Reviewer 1 Report
Dear Authors
I congratulate you for the extensive revision you have implemented in the current version of the manuscript water-2427621. I have seen that the observations reported by the reviewers in the first round of revision have been implemented. In the current version, I have not observed any other improvement that could be pointed out that would substantially increase the quality of the manuscript. I believe that it is in a condition to be considered for publication.
May the Lord be with you
Reviewer
Author Response
Response to Reviewer 1 Comments
Thank you for your valuable comments on the paper, we will continue to work according to your opinions in the future research. At the same time, I also wish you a smooth sailing on the road of scientific research in the future.

Reviewer 2 Report
I thank the Authors for kindly considering my comments.
However, my request in point 2 remains unresolved. I consider the repeated references to alleged links between the improvements observed over the years and the actions undertaken to be irrelevant. These sentences would make scientific sense only if an analysis of the various interventions implemented to improve the health of the system had been made in the paper and it had been demonstrated how, each intervention was followed by an improvement in the proposed indicators. Since this was not the purpose of the Author’s work, there is no need to put a link more than once that doesn't really say anything.
Based on the answer provided to me by the Authors in point 2, my suggestion is to saying, only once and in the concluding section, that the observation of the trend of the indicators can be useful for guiding the choices of any future planning actions.
Recommendation
For these reasons, the manuscript ver.2 should be accepted only after a minor revision
Author Response
Response to Reviewer 2 Comments
Thank you very much for taking the time to revise our paper. We have given all the answers to your questions.The contents are as follows :
Point 1:I consider the repeated references to alleged links between the improvements observed over the years and the actions undertaken to be irrelevant. These sentences would make scientific sense only if an analysis of the various interventions implemented to improve the health of the system had been made in the paper and it had been demonstrated how, each intervention was followed by an improvement in the proposed indicators. Since this was not the purpose of the Author’s work, there is no need to put a link more than once that doesn't really say anything.
Response 1:Thank you very much for taking the time to modify our paper, we have made changes to your questions, as follows :
(lines 318-319,320-321) Corresponding to the health trend is the proportion of river length above class III water quality ( C4 ), and the health degree of C4 in Zhengbianluo area is gradually developing towards a healthy trend. The discharge of human sewage discharge ( C1 ) has caused a certain degree of pollution to water quality. However, due to the government 's introduction of some water resources protection policies, the Zhengbianluo area ( C1 ) has gradually developed towards a better trend.
(lines 369-370) In the time dimension, the water cycle level in the Zheng-Bian-Luo region decreased from 2011 to 2015, with a major recovery from 2015 to 2021. and there is no obvious difference in the rate of improvement in the Zheng-Bian-Luo area. The water cycle health level of the Zheng-Bian-Luo area has gradually improved from the initial subpathological state to a healthy state.
(lines 384-385) In terms of the spatial dimension, the water cycle health level of the Zheng-Bian-Luo area is better, and the three cities reached a healthy (grade I) level by 2021, and the spatial difference is not significant. Water cycle health in the Zheng-Bian-Luo region was submorbid in 2011. In 2013, the health of the water cycle in the Zheng-Bian-Luo area became more serious.The three cities' water health has progressively improved since 2016, but the impact is not yet readily apparent. The three cities' water cycle health has greatly improved in recent years (from 2019 to 2021) .The spatial and temporal distribution map of water cycle health in the Zheng-Bian-Luo area can be seen in Figure 8.
(lines 506-507) Between 2011 and 2021, the Zheng-Bian-Luo region's water cycle health status was expected to increase significantly. In the Zheng-Bian-Luo region, the average comprehensive evaluation value, (distance from ideal solution), decreased from 0.447 in 2011 to 0.00 in 2021 (distance from perfect solution is 0, which is the ideal solution). There are some differences in the health level and improvement rate of water cycle between different regions, but the overall comprehensive evaluation value has been improved after 2014, indicating that all regions have adopted a series of measures suitable for local development according to their own actual situation, reflecting the observation of the trend of indicators. It is useful for the choice of future planning actions, and also reflects the importance of human beings to the healthy development of water cycle.
